# PSMA PET/CT in Renal Cell Carcinoma: An Overview of Current Literature

**DOI:** 10.3390/jcm11071829

**Published:** 2022-03-25

**Authors:** Stijn Muselaers, Selcuk Erdem, Riccardo Bertolo, Alexandre Ingels, Önder Kara, Nicola Pavan, Eduard Roussel, Angela Pecoraro, Michele Marchioni, Umberto Carbonara, Laura Marandino, Daniele Amparore, Riccardo Campi

**Affiliations:** 1Department of Urology, Radboud University Medical Center, 6525 GA Nijmegen, The Netherlands; 2Division of Urologic Oncology, Department of Urology, Faculty of Medicine, Istanbul University, Istanbul 34452, Turkey; erdemselcuk1@gmail.com; 3Department of Urology, San Carlo Di Nancy Hospital, 00165 Rome, Italy; riccardobertolo@hotmail.it; 4Department of Urology, University Hospital Henri Mondor, APHP, 94000 Créteil, France; alexandre.ingels@gmail.com; 5Department of Urology, School of Medicine, Kocaeli University, İzmit 41001, Turkey; onerkara@yahoo.com; 6Urology Clinic, Department of Medical, Surgical and Health Science, University of Trieste, 34127 Trieste, Italy; nicpavan@gmail.com; 7Department of Urology, University Hospitals Leuven, 3000 Leuven, Belgium; eduard.roussel@uzleuven.be; 8Department of Urology, San Luigi Gonzaga Hospital, University of Turin, Orbassano, 10124 Turin, Italy; pecoraro416@gmail.com (A.P.); danieleamparore@hotmail.it (D.A.); 9Department of Medical, Oral and Biotechnological Sciences, G. d’Annunzio University of Chieti, 66100 Chieti, Italy; mic.marchioni@gmail.com; 10Department of Emergency and Organ Transplantation-Urology, Andrology and Kidney Transplantation, University of Bari, 70121 Bari, Italy; u.carbonara@gmail.com; 11Department of Medical Oncology, IRCCS Ospedale San Raffaele, 20132 Milan, Italy; laura.lmarandino@gmail.com; 12Unit of Urological Robotic Surgery and Renal Transplantation, Careggi Hospital, University of Florence, 50121 Florence, Italy; riccardo.campi@unifi.it

**Keywords:** PSMA PET/CT, renal cell carcinoma, kidney cancer, imaging

## Abstract

Although the vast majority of prostate-specific membrane antigen (PSMA) positron emission tomography (PET) imaging occurs in the field of prostate cancer, PSMA is also highly expressed on the cell surface of the microvasculature of several other solid tumors, including renal cell carcinoma (RCC). This makes it a potentially interesting imaging target for the staging and monitoring of RCC. The objective of this review is to provide an overview of the current evidence regarding the use of PSMA PET/Computed Tomography in RCC patients.

## 1. Introduction

Renal cell carcinoma (RCC) is the most common solid tumor within the kidney and accounts for approximately 3% of all malignancies in the world [1]. RCC is not a single entity, but rather a collection of different types of tumors, each derived from the various parts of the nephron. RCC subtypes are all characterized by distinct molecular changes, histological features, and clinical phenotypes [2].

Due to the wide adoption of imaging modalities in clinical practice, the incidental finding of renal masses is rapidly increasing. One of the most difficult challenges in the diagnosis of RCC is that conventional imaging studies such as ultrasound and Computed Tomography (CT) cannot reliably distinguish benign solid lesions from RCC. Moreover, the management of RCC is heavily dependent on disease stage. Surgical resection is usually the first choice in the case of localized disease; however, approximately 30% of patients present with metastatic disease, and recurrence occurs in about 40% of patients treated for a localized tumor [3,4]. Therefore, the adequate characterization of suspect lesions based on imaging is essential to avoid invasive biopsies and superfluous surgery, both in localized and advanced disease.

In recent years, the role of prostate-specific membrane antigen (PSMA) PET/CT imaging has been deeply investigated in the staging of primary, biochemically recurrent, and metastatic prostate cancer [5]. However, despite its name, PSMA is not exclusively expressed in prostate cancer. Several other solid tumor types including RCC are known to show an overexpression of this antigen, particularly in endothelial cells of tumor-associated neovasculature [6,7,8].

In this literature review, we aim to provide an overview of the current evidence regarding the use of PSMA PET/CT in RCC patients.

## 2. Methods

A literature search was performed for studies on PSMA PET/CT in patients with RCC up to December 2021. MEDLINE databases (Pubmed and Web of Science) were searched using the following keywords: “Renal Cell Carcinoma” AND “PET/CT”, “Renal Cancer” AND “PET/CT”, “renal cancer” AND “PSMA PET/CT”, and “renal cell carcinoma” AND “PSMA PET/CT”. Although numerous case reports were found, mostly larger cohort studies and case series were selected for this review. Finally, references of articles found in the literature search were examined to find additional reports that met the scope of this review. An overview of the included literature on PSMA PET in RCC is provided in Table 1.

## 3. PSMA and PSMA PET/CT

PSMA is a type II transmembrane glycoprotein encoded by the FOLH1 (Folate hydrolase) gene and was first found as the target for monoclonal antibody 7EII-C5.3 in a preclinical study with prostate cancer cell lines. [9] Because of the high expression of this antigen in prostate cancer, it was quickly recognized that it could serve as an excellent target for both imaging and therapeutic approaches in this disease. In recent years, research in this field has experienced a huge surge, and there is a growing body of literature regarding radionuclide imaging and radioligand therapy with tracers targeting PSMA [10]. Currently, the most intensively studied and widely used anti-PSMA PET tracers are [^68^Ga] Ga-PSMA-HBED-CC and ^18^F-DCFPyL, although in recent years, alternative tracers, such as ^18^F-PSMA-1007, are emerging [11,12].

Gallium-68 (^68^Ga)-labeled PSMA-HBED-CC PET (also known as ^68^Ga-PSMA-11 or ^68^Ga-PSMA) was first described in preclinical studies in prostate cancer models [13]. Since then, a large amount of evidence supporting the role of ^68^Ga-PSMA PET imaging in prostate cancer has been published, summarized in a recent systematic review and meta-analysis [14]. Gallium-68 has a physical half-life of 67.71 min and is produced with a dedicated and costly generator that extracts the radionuclide from a source of decaying germanium-68.

In search of alternative anti-PSMA tracers, Chen et al. synthesized and evaluated in vivo fluorine-18 (^18^F)-labeled DCFPyL. Fluorine-18 has several important advantages such as the nearly pure positron emission, ease of radiolabeling, and relatively long physical half-life (110 min), which enables the radionuclide to be transported some distance from its point of production [15]. Since the initial studies, ^18^F-DCFPyL has become a widely used tracer in prostate cancer imaging [16,17,18].

In addition to the overexpression of PSMA in prostate cancer, several ex vivo studies have shown that other solid tumor types overexpress this antigen, particularly in endothelial cells of the tumor-associated neovasculature [6,7,8]. In the largest cohort to date, Spatz et al. retrospectively reported on 257 RCC patients (including clear cell, papillary, and chromophobe subtypes). Interestingly, stronger PSMA expression patterns seemed to correlate with higher grade, more advanced tumors, and poorer overall survival rates [8]. The overexpression of this antigen in several subtypes of RCC has led to the growing interest in the use of this target for the imaging of RCC.

## 4. PSMA PET/CT in Clear Cell Renal Cell Carcinoma

In 2014, the first case in which ^68^Ga-PSMA PET/CT was used for the diagnosis of metastatic ccRCC was reported by Demirci et al. [19]. Authors reported on a woman whose tumor was evaluated with both [^18^F] FDG PET/CT and ^68^Ga-PSMA PET/CT, and markedly more metastatic lesions were detected with the latter.

Rowe et al. described the first series in which the utility of ^18^F-DCFPyL PET in five patients with metastatic ccRCC was assessed. More metastatic lesions were detected with ^18^F-DCFPyL PET than with conventional imaging, demonstrating higher sensitivity (95% vs. 79%, respectively) [20]. In 2016, the same group performed an intrapatient comparison of FDG PET/CT and ^18^F-DCFPyL PET/CT in a patient with metastatic RCC (mRCC). The ^18^F-DCFPyL tracer demonstrated improved sensitivity in detecting small lesions and higher tracer uptake in these lesions [21]. However, an important limitation of both studies is the absence of histological confirmation. This limitation was overcome in another study of this group, where a moribund 52-year-old man with metastatic ccRCC after multiple lines of systemic therapy consented with ^18^F-DCFPyL PET/CT imaging shortly prior to his passing [22]. CT imaging demonstrated 55 sites of metastatic ccRCC, and in 54 out 55 (98.1%) of these lesions, uptake of the PSMA tracer was observed. Moreover, twelve additional lesions were found with ^18^F-DCFPyL PET, of which eight were accessible for biopsy. In seven out of these eight lesions (87.5%), ccRCC was histologically confirmed, while in the remaining (bone) lesion, there was a strong suspicion of a non-representative biopsy [22].

Another intrapatient comparative study was performed by Liu et al. [23]. In this study, in 15 patients with a history of ccRCC, a total of 36 lesions were detected by either ^18^F-DCFPyL PET/CT or ^18^F-FDG PET/CT. Histopathological confirmation was possible in some, but not all cases. A higher detection rate of soft tissue lesions was found upon using ^18^F-DCFPyL PET/CT, although this was not statistically significant (*p* = 0.125). The ^18^F-DCFPyL PET/CT tracer was found to be superior for the detection of bone lesions (*p* = 0.002). Imaging with ^18^F-DCFPyL PET showed significantly higher maximum standardized uptake values (SUVmax) and higher tumor-to-background ratios than ^18^F-FDG PET.

Rhee et al. reported on ^68^Ga-PSMA PET/CT in patients with newly diagnosed renal tumors and suspicion for metastatic disease on standard imaging (CT or MRI) [24,25]. Although the majority of patients had ccRCC, patients with papillary and unclassified RCC were also included in this analysis. Patients underwent biopsies of several suspicious lesions, and histopathological findings were correlated with those from the various imaging modalities. Primary lesions were PSMA-avid in all patients, and for metastases, PSMA PET/CT showed a superior sensitivity of 92.11% and a positive predictive value (PPV) of 97.22% when compared to conventional imaging (sensitivity 68.6%, PPV 80%).

The group of Sawicki et al. also identified the high accuracy of ^68^Ga-PSMA PET/CT for detecting RCC metastases [26]. This group evaluated ^68^Ga-PSMA PET/CT in five primary RCCs and 16 metastases in a total of six patients and correlated outcomes with histopathology. The ^68^Ga-PSMA PET/CT tracer was able to accurately detect RCC metastases, but the visualization of primary RCCs was hampered due to the high physiological ^68^Ga-PSMA uptake in the surrounding renal tissue, resulting in a poor tumor-to-background ratio. The mean difference in SUVmax between tumor and background was only 0.2 ± 0.3 (range 0.02–0.7). The authors therefore highlight that despite promising results in the detection of RCC metastases, the diagnostic value of ^68^ Ga-PSMA PET/CT in primary tumors seems to be limited.

Another prospective study that looked into the performance of ^18^F-DCFPyL PET/CT found 29 PSMA-avid lesions in 14 patients, whereas conventional imaging revealed only 21 metastases. [27] The three primary tumors in this cohort were detected with both imaging modalities. Because of the additional lesions found with ^18^F-DCFPyL PET/CT, over 20% of patients were no longer eligible for metastasis-directed therapy. In these cases, the change in clinical management was attributed to the use of PSMA PET imaging.

The group of Raveenthiran et al. reported changes in clinical decision making due to PSMA PET/CT outcomes in a larger, retrospective cohort of 38 patients. A change in clinical management due to ^68^Ga-PSMA PET/CT was found in 43.8% of primary staging scans and 40.9% of restaging scans, resulting in a total of 42.1% of all RCC cases [28]. Unfortunately, histological confirmation was not available for all additional lesions found with PET/CT, and therefore, the PPV could not be calculated in this study.

In the largest study with histopathological confirmation to date, Gao et al. retrospectively examined data of 36 patients with primary ccRCC. Preoperative ^68^Ga-PSMA PET/CT imaging was shown to correlate with pT stage, ISUP grade, and adverse pathology characteristics such as necrosis, or rhabdoid or sarcomatoid dedifferentiation [29]. Günhe et al. looked into the correlation of PSMA PET results and antigen expression in metastatic ccRCC. PSMA expression was confirmed with immunohistochemistry in all metastatic lesions, although intensity and distribution did not correlate with PET parameters [30]. The authors of this study concluded that despite the value of PSMA PET in the clinical evaluation of these patients, it cannot reliably predict histologic features of metastases.

In the most recent report to date, Tariq et al. describe a series of 11 patients in which an intrapatient comparison of PSMA and FDG PET/CT was performed. Patients in this study underwent dual tracer PSMA and FDG PET/CT after standard CT imaging. Overall, PET imaging was found to be more accurate than conventional imaging in RCC patients. In three patients, PET imaging had an important impact on clinical decision making. Concordant FDG and PSMA uptake in metastatic RCC lesions was observed in 82% of cases, with the remaining two cases showing discordant uptake favoring PSMA [31].

## 5. PSMA PET/CT in Non-Clear Cell Renal Cell Carcinoma

Several studies report the inclusion of patients affected by non-ccRCC [24,26,28,31,32], but larger case series regarding the performance of PSMA PET/CT in these patients remains very limited. Only one series has been published to date in which patients with papillary RCC (*n* = 3), chromophobe RCC (*n* = 2), unclassified RCC (*n* = 2), and Xp11 translocation RCC (*n* = 1) were included [33]. In this study, 73 metastatic lesions and 3 primary tumors were detected with conventional imaging. No additional lesions were identified with ^18^F-DCFPyL PET without a corresponding finding on conventional imaging. Only 10 out of the 73 detected metastases (13.7%) showed clear PSMA tracer uptake (median SUVmax = 3.25, range = 1.2–9.5). In total, 14 lesions (19.2%) had equivocal tracer uptake (median SUVmax = 2.85, range = 0.5–6.5), and 49 lesions (67.1%) showed no uptake greater than the background (median SUVmax = 1.7, range 0.2–3.0). The three primary renal tumors actually had a lower tracer uptake than the background with a tumor-to-background ratio ranging from 0.1 to 0.3. This is in line with the findings in the cohort of Sawicki et al., where the SUVmax in patients with a primary papillary or chromophobe tumor was markedly lower than in patients with the ccRCC subtype [26]. These results suggest that PSMA PET is less suitable for imaging of non-ccRCC subtypes, in both the primary and metastatic setting.

## 6. Therapy Response Monitoring

As one might expect, the literature about therapy response monitoring with PSMA PET/CT in RCC is also scarce. Mittlmeier et al. reported on ^18^F-PSMA-1007 PET in a response evaluation of both tyrosine kinase inhibitors (TKI) and immune checkpoint inhibitors (ICI) [31]. The ^18^F-PSMA-1007 PET tracer, along with conventional CT scanning, was used in mRCC patients prior to the initiation of systemic treatment and was repeated 8 weeks after therapy initiation. Overall, 11 patients with mRCC (eight ccRCC, two papillary RCC and one unclassified RCC) undergoing systemic treatment were included, with 7/11 receiving TKI treatment and 4/11 receiving CI. Concordant results between PSMA PET and CT after 8 weeks of treatment were only observed in 2/11 patients, and in the majority of cases, PSMA PET results indicated a partial or complete response, whereas the CT indicated stable disease. The authors hypothesize that ^18^F-PSMA-1007 PET may be able to assess treatment response on a molecular level earlier than morphological changes appear on CT imaging. According to this study, combined PET and conventional CT may provide complementary information for response assessment in mRCC patients undergoing systemic treatment, although this analysis is hampered by a rather small and heterogeneous study population.

Additional data regarding the use of PSMA PET for response monitoring are provided by the group of Siva et al., who reported on ^68^Ga-PSMA PET after stereotactic radiotherapy [32]. In this cohort of seven ccRCC patients and one papillary RCC patient, uptake of the PSMA ligand was typically more intense than the uptake of FDG PET. In addition, metabolic changes could be observed with both imaging modalities, although a more rapid response was observed with FDG PET. In line with the results of the group of Mittlmeier et al., PSMA PET could demonstrate in a response to treatment earlier than morphological appearances on conventional imaging. However, it is not clear whether PSMA PET has clinical benefit over FDG PET, which in any case is not recommended for routine use in RCC.

## 7. Future Perspectives of PSMA PET in RCC

According to currently available data, PSMA imaging in RCC holds promise for the future. However, it is not yet clear exactly where its utility lies in the clinical management of RCC.

One of the largest studies to date found that PSMA PET/CT may help to predict pT stage, ISUP grade, and adverse pathology in localized ccRCC [29], but it is uncertain whether this could influence clinical decision making, as these patients will most likely undergo surgery or ablative procedures. Moreover, several other studies indicate that primary RCCs (of all subtypes, including ccRCC) show varying PSMA avidity and low uptake of the tracer relative to the surrounding tissue [25,26,33]. Consequently, this low tumor-to-background ratio hampers local staging of RCC, which limits the added value of PSMA PET in localized disease. There are new PSMA tracers available that have no or less renal excretion, such as PSMA-1007. These tracers result in better tumor-to-background ratios and are likely more suitable for imaging primary tumors [34]. Alternatively, tracers that target other tumor-associated antigens such as carbonic anhydrase IX may be superior this context [35,36,37,38].

For the imaging of metastatic ccRCC, PSMA PET seems to hold the most promise, as the tracer uptake is markedly lower in the non-ccRCC subtypes [25,26,28,33]. Even though no firm conclusions can be drawn due to the limited degree of evidence, in ccRCC, PSMA PET consistently outperformed conventional imaging modalities such as CT, MRI, and FDG PET in various series and case reports. This is in concordance with immunohistochemistry studies, where the ccRCC subtype had the highest percentage of PSMA-positive tumors and also the highest PSMA staining intensity when compared to other subtypes [8]. As several studies show, staging and restaging with PSMA-targeted PET/CT may assist in clinical decision making and in selecting patients for metastasis-directed therapy [27,28]. Furthermore, as demonstrated by Mittlmeier et al., PSMA PET may play a role in early response monitoring of TKI and ICI treatment [31]. Intriguingly, changes in the intensity of PSMA uptake during systemic therapy might provide early response assessment or novel insight into the biological responses to treatment. With the earlier detection of progressive disease, unnecessary exposure to ineffective and expensive treatment might be avoided in these patients.

Besides response monitoring in mRCC patients, some groups go even further by suggesting the use of the PSMA antigen as a target for radioligand therapy with lutetium-177-labeled PSMA ligands [8,32,39]. Although this is a fascinating concept, it is as yet unclear which patients should be selected for such an approach. To date, no clinical trials with this intent have been initiated.

To conclude, PSMA PET is an exciting new tool for clinicians treating RCC patients. However, further research is warranted to define its exact role in the staging and restaging of all subtypes of RCC and its reliability in response monitoring of systemic treatment.

## Table

**Table 1 jcm-11-01829-t001:** Literature overview on PSMA PET in renal cell carcinoma.

Study	Year	Type	Location	Objectives	Number of Patients	Histology	Radiotracer	Comperator	Main Findings
Demirci et al. [19]	2014	Retrospective	Germany	First report of ^68^Ga-PSMA PET/CT in ccRCC.	1	ccRCC	^68^Ga-PSMA-HBED-CC	None	Imaging of ccRCC is feasible with PSMA PET.
Rowe et al. [20]	2015	Prospective	USA	To evaluate ^18^F-DCFPyL in metastatic ccRCC.	5	ccRCC	^18^F-DCFPyL	CT or MRI	PSMA PET has a higher sensitivity for detecting ccRCC metastases than conventional imaging (94.7% versus 78.9% respectively).
Rhee et al. [25]	2016	Prospective	Australia	To evaluate ^68^Ga-PSMA PET/CT for the detection of RCC.	10	8 ccRCC, 1 papRCC, 1 unclassified RCC	^68^Ga-PSMA-HBED-CC	CT	PSMA PET/CT has a sensitivity of 92.1 % and a PPV 97.2 % for detecting metastatic RCC.
Gorin et al. [22]	2017	Prospective	USA	To evaluate the accuracy of PSMA PET by histologically sample PSMA PET-only detected lesions.	1	ccRCC	^18^F-DCFPyL	Histology	PSMA PET/CT is able to accurately detect RCC metastases.
Sawicki et al. [26]	2017	Retrospective	Germany	To evaluate the diagnostic potential of PET/CT using a ^68^Ga-PSMA PET in RCC.	6	4 ccRCC, 1 papRCC, 1 chromRCC	^68^Ga-PSMA-HBED-CC	Histology	PSMA PET/CT is able to detect RCC metastases, but does not have additional diagnostic value in assessing the primary tumor.
Yin et al. [34]	2018	Prospective	USA	To evaluate ^18^F-DCFPyL PET/CT for detection of metastatic lesions of non-clear cell RCC.	8	3 papRCC, 2 chromRCC, 2 unclassified RCC, 1 Xp11 translocation RCC	^18^F-DCFPyL	CT or MRI	PSMA-based PET is not suitable for imaging non-ccRCC subtypes as only a small amount of suspected metastatic lesions are PSMA avid.
Meyer et al. [27]	2019	Prospective	USA	To evaluate the clinical utility of ^18^F-DCFPyL PET/CT in RCC patients.	14	ccRCC	^18^F-DCFPyL	CT or MRI	PSMA PET had a higher detection rate for metastatic lesions than conventional imaging.
Raveenthiran et al. [28]	2019	Retrospective	Australia	To evaluate the clinical utility of ^68^Ga-PSMA PET/CT in RCC patients.	35	Predominantly ccRCC, 1 chromRCC	^68^Ga-PSMA-HBED-CC	CT	PSMA PET/CT directly changed management in 42.1% of the RCC cases.
Gao et al. [29]	2020	Retrospective	China	To evaluate the correlation between PSMA PET parameters and pathological characteristics in primary ccRCC.	36	ccRCC	^68^Ga-PSMA-HBED-CC	Histology	PSMA PET/CT can identify aggressive pathological features of ccRCC..
Liu et al. [23]	2020	Retrospective	China	To compare the diagnostic performance of ^18^F-DCFPyL and ^18^F-FDG PET/CT in the restaging of ccRCC.	15	ccRCC	^18^F-DCFPyL	^18^F-FDG PET	PSMA PET showed a higher SUVmax and higher tumor-to-background ratios than FDG PET in ccRCC patients.
Gühne et al. [30]	2021	Prospective	Germany	To evaluate PSMA PET/CT for the detection of metastatic recurrence of ccRCC.	9	ccRCC	^68^Ga-PSMA-HBED-CC	Histology	Imaging ccRCC is feasible with PSMA PET/CT, but cannot reliably predict histologic features of metastases.
Tariq et al. [31]	2021	Retrospective	Australia	To compare the diagnostic performance of PSMA and ^18^F-FDG PET/CT in ccRCC	11	10 ccRCC, 1 unclassified RCC	^68^Ga-PSMA-HBED-CC, ^18^F-PSMA-1007	CT	PET imaging was found to be more accurate than conventional imaging, with PSMA PET outperforming FDG PET.

Abbreviations: RCC = renal cell carcinoma, ccRCC = clear cell renal cell carcinoma, papRCC = papillary renal cell carcinoma, chromRCC = chromophobe renal cell carcinoma, PPV = positive predictive value.

## Data Availability

Not applicable.

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
