# Peer review of "PSMA PET/CT in Renal Cell Carcinoma: An Overview of Current Literature"

_jcm, 2022, doi:10.3390/jcm11071829_

Round 1

Reviewer 1 Report

I have previously reviewed this paper and concur with my prior comments (see below). The revisions made to the original version are appropriate and sound.

The authors provide a thorough review of PSMA-targeted imaging in renal cell carcinoma. The name “PSMA” is indeed a misnomer as emphasized by these authors, as PSMA is not “specific” for prostate cancer. Further attention to this misnomer is needed and these authors should be commended for
highlighting the important relevance of the misnomer in RCC. The authors appropriately allude to potential uses for PSMA-targeted imaging in RCC both for diagnostic and potential therapeutic uses.

Reviewer 2 Report

Well written manuscript with a nice review on PSMA PET in RCC. 

This manuscript is a resubmission of an earlier submission. The following is a list of the peer review reports and author responses from that submission.

Round 1

Reviewer 1 Report

The authors provide a thorough review of PSMA-targeted imaging in renal cell carcinoma. The name “PSMA” is indeed a misnomer as emphasized by these authors, as PSMA is not “specific” for prostate cancer. Further attention to this misnomer is needed and these authors should be commended for highlighting the important relevance of the misnomer in RCC. The authors appropriately allude to potential uses for PSMA-targeted imaging in RCC both for diagnostic and potential therapeutic uses.

Author Response

We like to thank the reviewer for the time taken to comment on our work.

Reviewer 2 Report

Very well written and nice review of the literature.  Clear layout and easy to follow format.  With the field of PSMA in RCC slowly gaining traction, this is will be welcomed. 

Only one minor correction. 

Page 2, line 88 - I believe that "Fluor-18" should be Fluorine-18 or 18F .  If so, please change. 

Author Response

The reviewer correctly points out that the correct spelling of the radionuclide is Fluorine-18. This has been changed in the revised manuscript.

Reviewer 3 Report

In this paper, the authors provide an overview of the current evidence regarding the use of PSMA 32 PET/Computed Tomography in RCC patients. There are some concerns.

The author should check whether all the related literature have been involved.  Some recent literature were missed, such as

  • The role of dual tracer PSMA and FDG PET/CT in renal cell carcinoma (RCC) compared to conventional imaging: A multi-institutional case series with intra-individual comparison
  • Application of 18 F Prostate-Specific Membrane Antigen Positron Emission Tomography/Computed Tomography in Monitoring Gastric Metastasis and Cancer Thrombi from Renal Cell Carcinoma

The author should provide tables to sum up the literature utilizing PSMA-PET CT imaging for detection of ccRCC and non-clear cell renal cell carcinoma respectively.

Author Response

We would like to thank the reviewer for the comments on our work. The reason that the suggested literature was not included in our manuscript is that it was not available at the time this review was prepared. The paper of Tariq et al. regarding intrapatient comparison of FDG PET and PSMA PET is indeed a very nice addition and has been added to the manuscript. We are well aware that several other case reports are available, but we specifically aimed to provide an overview of the highest level of evidence available. Therefore, mostly larger cohort studies and case series were selected for this review. As suggested by the reviewer, a comprehensive table of the literature on PSMA PET was included in our manuscript was provided (please see the attachment).
